

# Combining simulation modeling and stable isotope analyses to reconstruct the last known movements of one of Nature's giants

Clive N. Trueman[1], Andrew L. Jackson[2], Katharyn S. Chadwick[1], Ellen J. Coombs[3,4], Laura J. Feyrer[5], Sarah Magozzi[1,6], Richard C. Sabin[3] and Natalie Cooper[3]

[1] School of Ocean and Earth Science, University of Southampton, Southampton, United Kingdom
[2] School of Natural Sciences, University of Dublin, Trinity College, Dublin, Ireland
[3] Department of Life Sciences, Natural History Museum, London, United Kingdom
[4] Department of Earth Sciences, University College London, University of London, London, United Kingdom
[5] Department of Biology, Dalhousie University, Halifax, NS, Canada
[6] Department of Geology and Geophysics, University of Utah, Salt Lake City, UT, United States of America

Corresponding author
Natalie Cooper,
natalie.cooper@nhm.ac.uk

## ABSTRACT

The spatial ecology of rare, migratory oceanic animals is difficult to study directly. Where incremental tissues are available, their chemical composition can provide valuable indirect observations of movement and diet. Interpreting the chemical record in incremental tissues can be highly uncertain, however, as multiple mechanisms interact to produce the observed data. Simulation modeling is one approach for considering alternative hypotheses in ecology and can be used to consider the relative likelihood of obtaining an observed record under different combinations of ecological and environmental processes. Here we show how a simulation modeling approach can help to infer movement behaviour based on stable carbon isotope profiles measured in incremental baleen tissues of a blue whale (*Balaenoptera musculus*). The life history of this particular specimen, which stranded in 1891 in the UK, was selected as a case study due to its cultural significance as part of a permanent display at the Natural History Museum, London. We specifically tested whether measured variations in stable isotope compositions across the analysed baleen plate were more consistent with residency or latitudinal migrations. The measured isotopic record was most closely reproduced with a period of residency in sub-tropical waters for at least a full year followed by three repeated annual migrations between sub-tropical and high latitude regions. The latitudinal migration cycle was interrupted in the year prior to stranding, potentially implying pregnancy and weaning, but isotopic data alone cannot test this hypothesis. Simulation methods can help reveal movement information coded in the biochemical compositions of incremental tissues such as those archived in historic collections, and provides context and inferences that are useful for retrospective studies of animal movement, especially where other sources of individual movement data are sparse or challenging to validate.

## INTRODUCTION

Migratory species pose a particular challenge for conservation practitioners because their effective conservation relies on protection at many, often distant, sites (*Runge et al., 2014*). Migratory species may also be particularly vulnerable to changes in climate or human use of the environment, as they are influenced by conditions in multiple locations across different parts of their life cycle (*Robinson et al., 2009*). Identifying threats to migratory species, understanding species responses to global change and developing effective conservation measures all require information on the movements of individual animals over multiple years, ideally for both historic and present-day populations.

With the development of electronic tagging technology, studies of the distributions of animals have been supplemented by detailed information about the movement of individuals (*Holdo & Roach, 2013*), providing unprecedented levels of detail regarding individual movement behaviours and potential environmental drivers for movement. However, despite the tremendous advances made using direct telemetry devices, tags are expensive, difficult to deploy on large whales, and rarely report information for longer than six months (*Bailey et al., 2009*; *Best, Mate & Lagerquist, 2015*; *Mate, Mesecar & Lagerquist, 2007*). Data on individual-level, multi-annual movements remain scarce, especially for dispersed, wide-ranging and long-lived marine species such as baleen whales (Mysticeti; *Bailey et al., 2009*; *Hall-Aspland, Rogers & Canfield, 2005*; *Ryan et al., 2013*).

Natural tags, based on intrinsic biochemical information such as the stable isotope compositions of tissues, provide an important additional tool for investigations of individual animal movements (*Busquets-Vass et al., 2017*; *Hobson & Wassenaar, 2008*; *West et al., 2006*). Present-day populations can be studied with material collected in the field, while historic populations can be studied using samples taken from museum collections; rich archives of behavioral information that are often under-utilised (*Lister et al. , 2011*). The stable isotope composition of animal tissues reflects the isotopic composition of diet at the time and place of ingestion, integrated over the timescale of tissue growth (*Boeklen et al., 2011*). However, relating the isotopic compositions of marine animal tissues to the most likely location of tissue growth is complicated by sources of uncertainty including spatial variation in the isotopic composition of diet (the isotopic baseline; *McMahon et al., 2015*; *West et al., 2006*), and numerous biochemical factors associated with the fractionation of stable isotopes between predator tissues and prey (*Boeklen et al., 2011*; *Newsome, Clementz & Kock, 2010*). This uncertainty is compounded when multiple samples are taken through time in the same individual, as temporal variation in the isotopic baseline, rates of movement of the consumer across isotopic gradients, and growth rates of the sampled tissues must also be considered explicitly. The large number of uncertain variables complicates logic-based interpretation of sclerochemical data. Consequently, relatively little attention has been given to the potential to infer spatially-explicit individual movement histories from the isotopic compositions of sequentially sampled incrementally-grown tissues (*Trueman & St John Glew, 2019*).

Considering multiple, non-exclusive alternative hypotheses rather than binary exclusive hypotheses is recognized as an important but difficult barrier in scientific and ecological

reasoning (*Betini, Avgar & Fryxell, 2017*). Simulation modeling is one tool that can be used to generate predictions expected under multiple, user-defined hypothetical conditions. The results of simulation models can be compared to observed data to identify which of a suite of hypotheses are more consistent with observed data, and to determine the sensitivity of an observed outcome to assumptions inherent in a method. Recently, mechanistic computational models have been developed that provide isoscapes—relatively accurate predictions of isotopic variability at high temporal and spatial resolution. These models can be coupled to Lagrangian or agent-based models of animal movement to predict the range of isotopic compositions that could be recorded in animal tissues under differing movement scenarios (*Carpenter-Kling et al., 2019*; *Darnaude et al., 2014*; *Sakamoto et al., 2019*).

Here we develop a simulation modeling approach and use it to interpret stable isotope profiles measured in the baleen plate of a blue whale (*Balaenoptera musculus*). Mysticete whales are characterized by the development of baleen, keratinous structures in the upper jaw used to filter food items from seawater. Baleen is ideal for stable isotope studies because keratin grows continuously through an individual's life, and once laid down it is metabolically inert (*Best & Schell, 1996*; *Hobson & Schell, 1998*). Baleen plates therefore offer a continuous isotopic record of behavior typically reflecting multiple years of life of an individual whale. Baleen is worn away at the tips over time, so a baleen plate reflects the most recent years of life, and rarely records an individual's entire lifespan. Among the mysticete whales, blue whales are a particularly attractive target for model-based isotope movement work as they have a consistent, low trophic level diet and feed continuously through the year, likely driven by high energetic costs of maintaining extreme muscle mass (*Goldbogen et al., 2015*).

Similar to other large balaenopterid whales, blue whales are generally assumed to conduct annual migrations between high and low latitudes (*Hucke-Gaete et al., 2018*). Their migration routes and feeding areas are thought to be shaped by highly productive regions (*Branch et al., 2007*), however, substantial individual variation in movement history and migration patterns has been documented (*Busquets-Vass et al., 2017*). The northeast Atlantic population of blue whales is estimated to be small—only 1,000 individuals (*Pike et al., 2009*)—with foraging observed around Iceland and historically in the Norwegian and Barents Seas in summer months (*Pike et al., 2009*), and wintering in the upwelling systems between Mauritania and the Cape Verde Islands (*Baines & Reichelt, 2014*). Northward migrations may occur along mid-Atlantic corridors with peak sightings in the Azores in the spring (*Silva et al., 2013*), and southerly migrating blue whales are frequently detected in waters to the west of the UK between November and December (*Baines, Reichelt & Griffin, 2017*; *Charif & Clark, 2009*; *Reeves et al., 2004*; *Visser et al., 2011*).

The movements of blue whales in the northeast Atlantic can be generalized from disparate information sources but are poorly understood in detail. Movements of individual animals over timescales of a year or more, the degree of connectivity between summer and winter feeding areas, and the level of temporal consistency in movement behaviour within individuals represent large questions for understanding the ecology and life history of this species. Identifying the historical movement patterns of individual blue whales is important

for understanding the current context of whale migrations, and potential for human-whale conflicts as populations recover.

We aim to show that simulation modeling of stable carbon isotope tracers can be used to test hypotheses about individual-level movement behavior. We specifically ask whether the observed isotopic record is more consistent with migratory or resident behavior at annual timescales. We draw on newly developed predictions of spatio-temporal variation in phytoplankton and stable carbon isotope composition at global and monthly resolution (Fig. S1), and couple these with an agent-based model of idealized whale movement. By simulating the carbon isotope record expected for several differing, plausible hypotheses of movement behavior, we can assess the range of isotopic expression expected for each alternative hypothesis. Modeled baleen isotope records associated with each movement scenario are compared to the observations of carbon isotope variations found in the baleen of a culturally significant blue whale, and illustrate a conceptual approach for interpreting the life history information contained in incremental tissues. This study also provides new information and understanding of the possible movements of an individual blue whale that lived in the North Atlantic during the first era of industrial whaling.

# MATERIALS AND METHODS

## Stable isotope extractions from baleen

Baleen was collected from a mature female blue whale, which stranded off the coast of Wexford Ireland in March 1891. The whale "Hope" was estimated to be at least 15 years old when she died and is currently on display in the Natural History Museum (NHM), London (specimen NHMUK.1892.3.1.1). The specimen's baleen plate was cleaned with ethanol to remove surface contaminants such as skin/gum or other lipids that can influence isotopic signals. 1 mg samples of keratin powder were then collected from the plate using a hand-held drill and grinding bit. 97 samples were taken at 1 cm intervals, 0.5 cm from the outer edge of the plate, starting at the proximal (gingival) section that contains the most recent tissue. Baleen is assumed to grow at a relatively constant rate, so the samples are equally spaced through time (*Best & Schell, 1996*). Carbon and nitrogen isotope analysis was performed simultaneously via continuous-flow isotope ratio mass spectrometry at the University of Southampton SEAPORT Stable Isotope Ratio Mass Spectrometry Laboratory (Southampton, UK), using a Vario Isotope select elemental analyser, coupled to an Isoprime 100 isotope mass spectrometer. Replicates using internal laboratory standards (L-glutamic acid (C), Glutamic acid (CT standard), acetanilide and protein standard OAS) were used for quality control and calibration. C:N ratios for samples ranged from 3.28 to 3.72, well within the acceptable theoretical range for pure keratin ($3.4 \pm 0.5$) allowing for comparison among samples (*Hobson & Schell, 1998*). All data are available from the NHM Data Portal (*Trueman et al., 2018*; https://doi.org/10.5519/0093278).

## Time calibrating stable isotope profiles

Many mysticete whales are characterized by regular, seasonal variations in behavior with increased feeding rates, typically in more productive higher latitudes in summer and reduced feeding rates, often in warmer waters, in winter. The combination of movement
across isotopic gradients and physiological variations in feeding and excretion rates leads to marked, regular cyclicity in baleen stable isotope data. Assuming such cyclicity represents annual seasonal behavioural variations, the distance between successive isotopic peaks or troughs may indicate the growth rate of the baleen (*Schell, Saupe & Haubenstock, 1989*; *Best & Schell, 1996*; *Aguilar et al., 2014*; *Busquets-Vass et al., 2017*). Accordingly, baleen growth rates have been estimated from patterns of $\delta^{15}$N cyclicity in fin whales (averaging 20 cm y$^{-1}$, *Aguilar et al., 2014*) and blue whales (average growth rate estimated as $15.5 \pm 2.2$ cm y$^{-1}$, *Busquets-Vass et al., 2017*). We quantified the periodicity of $\delta^{15}$N fluctuations within Hope's baleen sample using Fourier Transform analysis (periodogram function in the R package TSA; *Chan & Ripley, 2012*; *Cardona et al., 2017*; Fig. S2), revealing an estimated growth rate of 13.5 cm y$^{-1}$, which is remarkably similar to the mean isotope-derived baleen growth rates for blue whales from the East Pacific of $15.5 \pm 2.2$ cm y$^{-1}$ (*Busquets-Vass et al., 2017*). We assumed a date for the most recent baleen sample as 1st March 1891, 24 days prior to the stranding date, 25th March 1891. Note that we make an explicit assumption of continuous growth in baleen within the growth of the individual, informed from the fluctuations in $\delta^{15}$N values. The assumption of constant growth (or the growth rate used) could be varied within a model simulation framework. We do not draw on $\delta^{15}$N values for any subsequent interpretations, avoiding conflating the signal used for time calibration with additional inferences.

## Modeled isoscapes

Isotope-enabled biogeochemical ocean models (*Magozzi et al., 2017*; *Schmittner & Somes, 2016*) were used to characterize the isotopic composition of phytoplankton expected in different potential foraging grounds (Fig. S1). $\delta^{13}$C POM values were simulated at 1° and monthly resolution using an isotopic extension to the NEMO-MEDUSA ocean biogeochemical model (*Magozzi et al., 2017*; *Yool, Popova & Anderson, 2013*). A full description of the carbon isotope model is provided by *Magozzi et al. (2017)*, but briefly $\delta^{13}$C POM values are estimated from modeled growth rates of silica limited (diatom) and non-silica limited plankton, concentrations of dissolved $CO_2$ and estimated $\delta^{13}$C values of dissolved inorganic carbonate and $CO_2$. We used monthly simulated $\delta^{13}$C values drawn from output from the NEMO-MEDUSA model system associated with climatological data from 2000 to 2010. As described in *Magozzi et al. (2017)*, using model outputs averaged across decadal scales to drive isotopic estimates reduces the chance of anomalous years influencing the predicted spatio-temporal patterns in phytoplankton $\delta^{13}$C values. We assume that the broad oceanographic distribution of $\delta^{13}$C POM values has remained similar between 1890 and 2000. The addition of fossil fuel derived carbon to the atmosphere and ocean has increased the concentration and reduced the $\delta^{13}$C values of dissolved inorganic carbon and POM (the Suess effect; *Young et al., 2013*). While absolute $\delta^{13}$C values have reduced since pre-industrial times, here we assume that latitudinal and seasonal variations in $\delta^{13}$C values have remained constant. We do not draw inferences about location from absolute $\delta^{13}$C values and thus the reduction in oceanic $\delta^{13}$C POM values does not directly influence inferences about location or movement. We are therefore

satisfied that the model can be applied to interpret historic baleen isotope data at least at ocean-basin scale spatial resolution, though we note that this is an assumption.

Annual average $\delta^{13}$C POM values largely vary with latitude, with lower values in more northerly regions. In the central North Atlantic, $\delta^{13}$C POM values are relatively high in the west, reflecting warm gulf stream waters (Fig. S1). The isotopic composition of carbon in phytoplankton also varies through seasons as isotopic fractionation of carbon during photosynthesis is strongly influenced by sea surface temperature (*Laws et al., 1995*; *Magozzi et al., 2017*). Thus temporal variations in $\delta^{13}$C POM values are superimposed on latitudinal gradients. The scale and nature of temporal variation in $\delta^{13}$C POM values also varies with latitude, with higher latitude seas showing greater intra-annual variation in $\delta^{13}$C POM values linked to strongly seasonal phytoplankton growth dynamics.

## Agent-based whale movement model (hypothesis generation)

We simulated the likely isotopic expression associated with alternative movement hypotheses by building an agent-based movement model in R (*R Core Team, 2018*), where movement behavior is influenced by sea surface temperature and phytoplankton biomass estimates provided by NEMO-MEDUSA (*Yool, Popova & Anderson, 2013*), and bathymetry from the General Bathymetric Chart of the Oceans (GEBCO; *Amante & Eakins, 2009*) extracted using the marmap R package (*Pante & Simon-Bouhet, 2013*; Fig. S3).

We tested two main hypotheses: whether the whale was resident or migratory, and further tested hypotheses associated with variation in the location of residency or starting location for migratory whales. We coded whale movements with the likelihood, direction, and extent of movement influenced by behavioral state, sea surface temperature, water depth, and phytoplankton concentration (as a proxy for zooplankton food availability). Movement was coded as a set of probabilistic rules, informed by the literature on blue whale behavior (e.g., *Wilson & Mittermeier, 2014*). All terms were expressed as probability distributions, yielding multiple potential movement tracks.

In the models, at each daily time step the likelihood of moving, the direction (north, south, east, west, northeast, northwest, southeast, or southwest) and the linear distance of movement, are all influenced by the following. (i) Behavioural state (migrating north, migrating south, or foraging, fixed according to month as defined by the operator). Here, northerly migrations were coded to occur in spring, and southerly migrations in autumn. Foraging was possible at any time of year, and was triggered when whales encountered high concentrations of plankton. (ii) Sea surface temperature (°C). When migrating north, whales were more likely to move towards lower temperatures provided they were above the minimum temperature threshold (3 °C), whereas whales migrating south sought warmer waters. (iii) Water depth (m; from GEBCO; *Amante & Eakins, 2009*). Whales were less likely to move into waters less than 400m deep, and increasingly unlikely to move into shallower waters. (iv) Phytoplankton concentration (mmol Nm$^{-3}$, for combined diatom and non-diatom communities; *Yool, Popova & Anderson, 2013*). This was included as a proxy for zooplankton food availability. Whales are more likely to move towards (or remain within) areas of high phytoplankton density, particularly during the foraging

behavioural state. At each daily step, the probability of movement, movement direction and distance traveled are sampled from probability distributions to allow individual variation.

Initial boundary conditions are defined with a maximum temperature of 25 °C and minimum temperature of 3 °C. The likelihood of movement (i.e., whether to move or not from the current location) is sampled from a binomial distribution with the probability of movement influenced by behavioural state and external conditions. The maximum daily movement distance permitted in each behavioural state is defined as a random sample of a Gaussian distribution with specified mean and standard deviation (see Table S1 and Fig. S3).

The agent-based model provides daily location data points that are then used to extract $\delta^{13}C$ values from the isotopic extension to the NEMO-MEDUSA model and assembled to build simulated isotopic profiles. We used $\delta^{13}C$ POM values modeled at monthly resolution to simulate the isotopic expression of phytoplankton expected to be encountered by whales exhibiting differing movement behaviours. The stable isotope compositions of keratin at a given point in the baleen will reflect the stable isotope compositions of the krill it was feeding on in the weeks prior to keratin growth (as a best guess). Assimilation of carbon into krill tissues will dampen the temporal variability seen in POM, effectively producing a temporal average over the timescale of isotopic turnover within krill. We roughly assume the rate of assimilation of phytoplankton biomass into krill to be complete between two and four months and therefore we resampled the $\delta^{13}C$ POM values in each one degree cell to reflect an average of isotopic compositions in phytoplankton in the two months prior to the sampling date. Average values were weighted according to the proportional plankton biomass estimated for each month.

Carbon isotope values are also likely to be altered during transfer from plankton to krill, as $^{12}C$ is preferentially lost through respiration. The degree of such trophic fractionation is unclear, however, and as we do not draw interpretations based on absolute $\delta^{13}C$ values, rather on the relative $\delta^{13}C$ values across the length of the baleen plate, we do not need to quantify this trophic enrichment effect. We assume that any trophic fractionation effect is constant in time and space. We also assume that carbon available for synthesis of keratin is drawn from carbon released by respiration of diet captured opportunistically throughout the year (*Bailey et al., 2009*; *Baines, Reichelt & Griffin, 2017*; *Branch et al., 2007*; *Busquets-Vass et al., 2017*; *Hucke-Gaete et al., 2018*; *Lesage et al., 2017*; *Silva et al., 2013*; *Visser et al., 2011*). For clarity, we chose to limit the number of free variables in the current study, but the fixed assumptions associated with the temporal lag between phytoplankton and baleen and trophic fractionation could be lifted and tested within the simulation framework we describe.

By manipulating the relative duration of the foraging and migratory behavioural states we produced simulated records of isotopic compositions of carbon expected under differing movement trajectories. We simulated the isotopic expression expected for (a) residency in each known hotspot for blue whale sightings or historic hunting grounds in the North Atlantic (Norwegian/Barents Sea, West Ireland, Canaries/Azores and Mid Atlantic Ridge, and the Cape Verde/Mauritanian upwelling area; *McDonald, Mesnick & Hildebrand, 2006*; *Reilly et al., 2008*; *Sigurjónsson, 1995*; Fig. S4); and (b) seasonal migration between

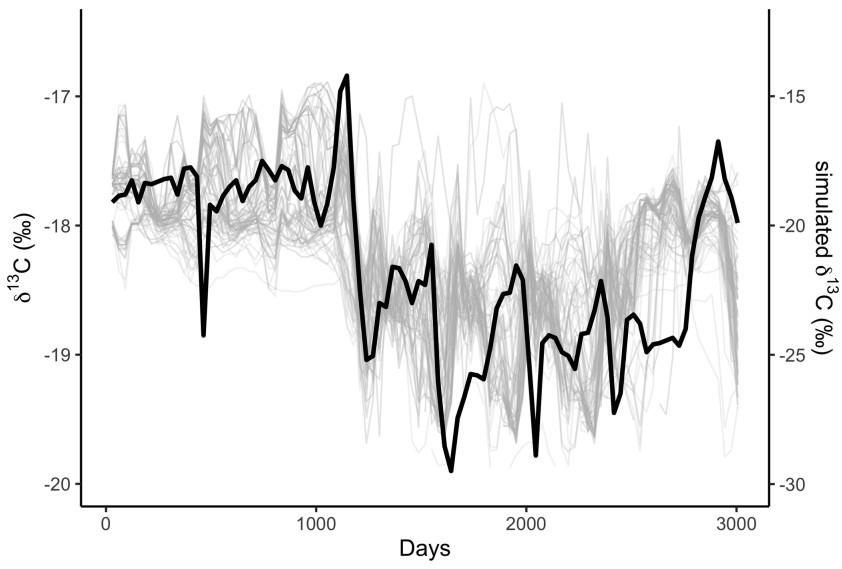

**Figure 1** **Correlations among simulated $\delta^{13}$C from the top 10% best fitting migratory movement models (grey lines, right hand *y*-axis) and $\delta^{13}$C from baleen (black line, left hand *y*-axis; see Fig. 2).** Simulated $\delta^{13}$C values are six month moving average values for the time series of simulated plankton $\delta^{13}$C values in that location, reflecting temporal integration of phytoplankton $\delta^{13}$C values within the food chain before ingestion by the whale as krill. The end points of the simulations and empirical data have been aligned to coincide.

high latitudes and subtropical latitudes. Within each movement hypothesis simulation, individual whales were parameterized with random variation in their maximum daily movement, providing a range of behaviours; a larger range of maximum daily movement was introduced in migratory whales.

We simulated two years of residency 30 times for each residency hotspot (Fig. S4). We simulated seven years of whale movements 1,200 times, then excluded simulations where the virtual whale movement model failed (i.e., the simulated whale became 'trapped' in coastal features) before reaching the 3,019 days of the baleen record, leaving 1,049 simulations. We then compared the simulated stable isotope profiles (Fig. 1) to the profile measured in the blue whale baleen (Fig. 2) with simple linear regressions.

R code for all analyses is available from GitHub (https://github.com/nhcooper123/blue-whale-bes; *Trueman, Jackson & Cooper, 2019*).

## RESULTS

### Model simulation results

We initially simulated temporal variations in baseline (phytoplankton) isotopic compositions that would be encountered by whales foraging within broad geographic regions (Norwegian/Barents Sea, West Ireland, Canaries/Azores and Mid Atlantic Ridge, and the Cape Verde/Mauritanian upwelling area Fig. S4). Strong seasonal dynamics in $\delta^{13}$C values are evident in northerly regions such as the Norwegian/Barents Sea, characterized by a rapid increase to annual maximum $\delta^{13}$C values associated with the onset of the spring

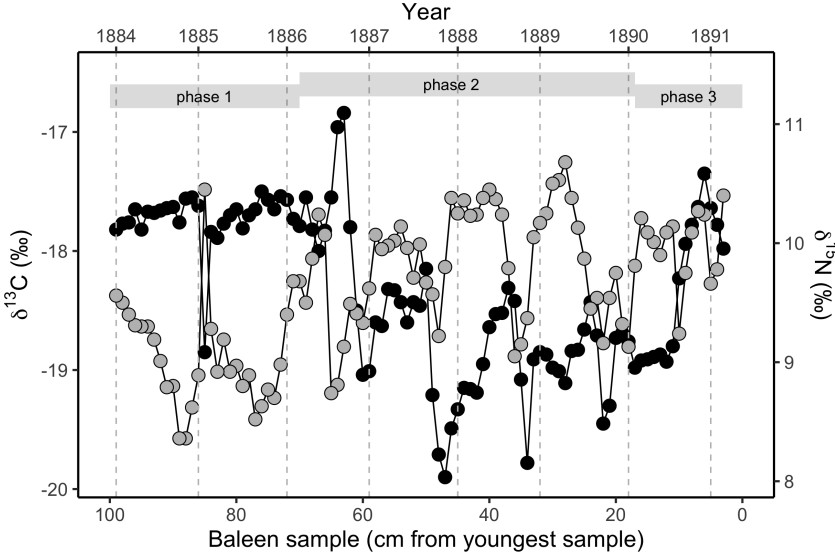

**Figure 2** **Variation in stable isotope values in the NHM blue whale, expressed as $\delta^{13}$C (black circles, left $y$-axis) and $\delta^{15}$N (grey circles, right $y$-axis).** Samples were taken longitudinally through the baleen plate ($n = 97$ samples from a single baleen plate for both isotopes). There is strong annual periodicity and cross-correlation (Figs. S2 and S5) in both isotopes. The approximate relationship to years assuming a growth rate of 13.5 cm y$^{-1}$ is shown on the upper $x$-axis, and year boundaries are indicated by vertical dotted grey lines. Phases are the three behavioural phases defined in the text.

phytoplankton bloom, followed by a more gradual decline in $\delta^{13}$C values towards minima in winter conditions (Fig. S4). In temperate latitudes west of the British Isles, seasonal dynamic cycles are present, but dampened (Fig. S4), whereas in sub-tropical regions exemplified by the Canaries, Mid Atlantic Ridge and particularly Mauritanian upwelling areas, $\delta^{13}$C values are relatively high and constant throughout the year (Fig. S4).

Adding seasonal north-south migrations within mid-high latitude regions to foraging models yielded simulated profiles with regular isotopic fluctuations of relatively high amplitude reflecting $\delta^{13}$C minima preceding the June bloom (Fig. S4, Fig. 2).

## Atlantic whale baleen measured isotopic record

The baleen plate yielded 97 discrete samples of baleen. $\delta^{15}$N values measured in the baleen plate display regular cyclical fluctuations throughout the length of the plate. Fourier analysis (*Cardona et al., 2017*) revealed a strong periodic repetition with a 13.3 cm periodicity, with a mean spacing of 13.5 cm, assumed to represent annual periodicity. Therefore, given the date of stranding (25th March 1891), and estimated baleen growth rates of 13.5 cm y$^{-1}$, we reconstructed a timeline for $\delta^{13}$C and $\delta^{15}$N fluctuations in the baleen over seven full years of the whale's life (early 1884–spring 1891).

$\delta^{13}$C values are relatively high and constant in the oldest (most distal) 35 cm of the baleen plate, associated with a weakening of the periodic fluctuation seen in $\delta^{15}$N values, and an overall reduction in $\delta^{15}$N values. This is followed by a clear change towards repeated fluctuations in $\delta^{13}$C values in the middle 50 cm of the baleen plate, with a similar periodicity of 13.5 cm (Fig. 2; Fig. S2). A second, distinct change in the pattern of $\delta^{13}$C values along the

baleen plate occurs at around 26 cm from the gingival end, with a transition to uniform, relatively low $\delta^{13}$C values followed by an abrupt transition to positive $\delta^{13}$C values and a decline in $\delta^{13}$C values in the most recent (most gingival) 3 cm of the record.

The $\delta^{13}$C and $\delta^{15}$N profiles therefore divide the isotopic record into three distinct phases that we assume reflect changes in the whale's behaviour. In behavioural phase one (from the start of the record to spring 1886), we find relatively stable, elevated $\delta^{13}$C values, and relatively low $\delta^{15}$N values (Fig. 2; Figs. S2 and S5). In behavioural phase two (summer 1886 to spring 1890) $\delta^{15}$N values are relatively high and $\delta^{13}$C values are relatively low with coincident cyclical fluctuations in both $\delta^{13}$C and $\delta^{15}$N values. In the last year of life the cyclical pattern is disrupted, with constant low $\delta^{13}$C values for approximately six months in the first half of 1890, before a rapid switch to relatively high $\delta^{13}$C values in the second half of 1890. The final three months of the record (behavioural phase three) show a progressive fall in $\delta^{13}$C values (Fig. 2; Figs. S2 and S5). Cross-correlation analysis demonstrates a strong negative covariance between $\delta^{13}$C and $\delta^{15}$N values within behavioural phase two (Fig. 2; Fig. S5), but no relationship between $\delta^{13}$C and $\delta^{15}$N values exists during behavioural phase one.

## Model—measured comparison results

For each of the three behavioural phases identified within the baleen plate, the modeled simulation results were compared to the observed stable isotope profile. In the observed profile, behavioural phase one is characterised by relatively high and constant $\delta^{13}$C values. $\delta^{15}$N values in this phase are relatively low and the isotopic cyclicity is absent (Fig. S2). The relatively high and seasonally-invariant $\delta^{13}$C values seen during behavioural phase one were reproduced only in simulations of whales resident in subtropical areas of the North Atlantic. Our simulations identify a range of possible locations for the whale (Fig. 3), although areas around the Mauritanian coast and Cape Verde Islands, a known current and historic winter feeding area for blue whales (*Baines & Reichelt, 2014*; *Reeves et al., 2004*), and potentially to the west of the Azores (Fig. 3), most closely match the measured profile. Temporal dynamics in $\delta^{13}$C values observed in the measured baleen during phase two cannot be reproduced in the simulated resident whales (Fig. 1, Fig. S4). During behavioral phase two, the observed low $\delta^{13}$C values likely imply foraging in colder, more northerly latitudes.

To further examine this, we modeled 3.5 years of seasonal north-south migration in the northeast northern Atlantic. We simulated 1,200 individual movement patterns, and compared the simulated baleen $\delta^{13}$C records of 1,049 runs yielding full time series records to the measured records with simple linear regressions. Simulated baleen $\delta^{13}$C profiles produce a good fit to measured profiles, the median $r^2$ value across 1,049 simulated profiles was 0.31, and the maximum was 0.65 (Fig. S6). The top 10% best fitting simulated profiles are shown in Fig. 3. Behavioral phase two is best simulated by seasonal migrations between summer foraging in northern areas (e.g., Norwegian Sea/Barents Sea/Iceland region), and winter foraging in a broad region between the UK and more southerly, subtropical waters. Better-fitting models in general were those predicting a greater latitudinal foraging range, and foraging in more northerly waters (Figs. S7 and S8). Best-fit model distributions are
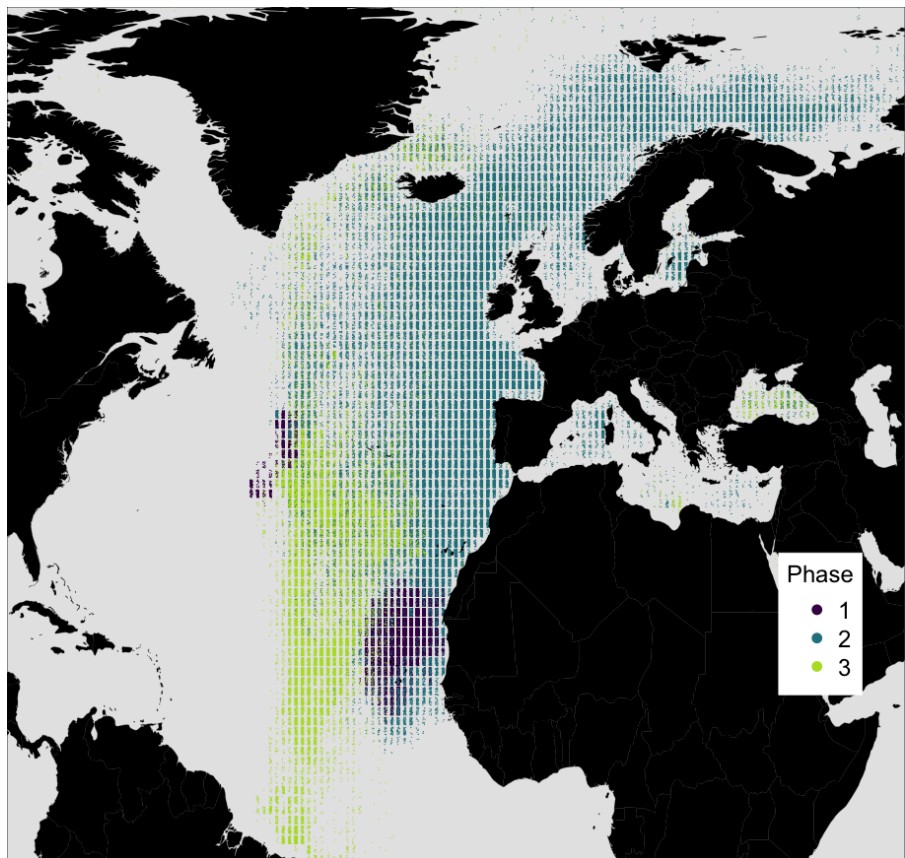

**Figure 3** **Simulated locations of the whale taken from the top 10% best fitting migratory movement models.** Colours reflect the behavioural phase. Phase one is early 1884 to spring 1886, phase two is summer 1886 to spring 1890, and phase three is spring 1890 to spring 1891.

largely consistent with current understanding of blue whale distributions in the northeast Atlantic (*Baines, Reichelt & Griffin, 2017*; *Baines & Reichelt, 2014*; *Reeves et al., 2004*), with perhaps greater importance of winter foraging in temperate regions (Fig. 4).

## DISCUSSION

Long-term, multi-annual data on the movement patterns and reproductive ecology of individual migratory marine animals are scarce. Stable isotope compositions of incrementally-grown tissues provide a promising source of additional indirect information regarding spatio-temporal behavior, but interpreting such profiles is difficult.

Here we show how numerical models can be used to simulate the temporal isotopic expressions expected within incrementally grown tissues associated with contrasting defined spatio-temporal foraging behaviours. We tested whether the isotopic variations observed across 6–7 years of growth of a baleen plate in a single blue whale in the northeast Atlantic could be better simulated by agent-based models coding for year round residency or latitudinal migration. The isotopic compositions in baleen from Hope showed three

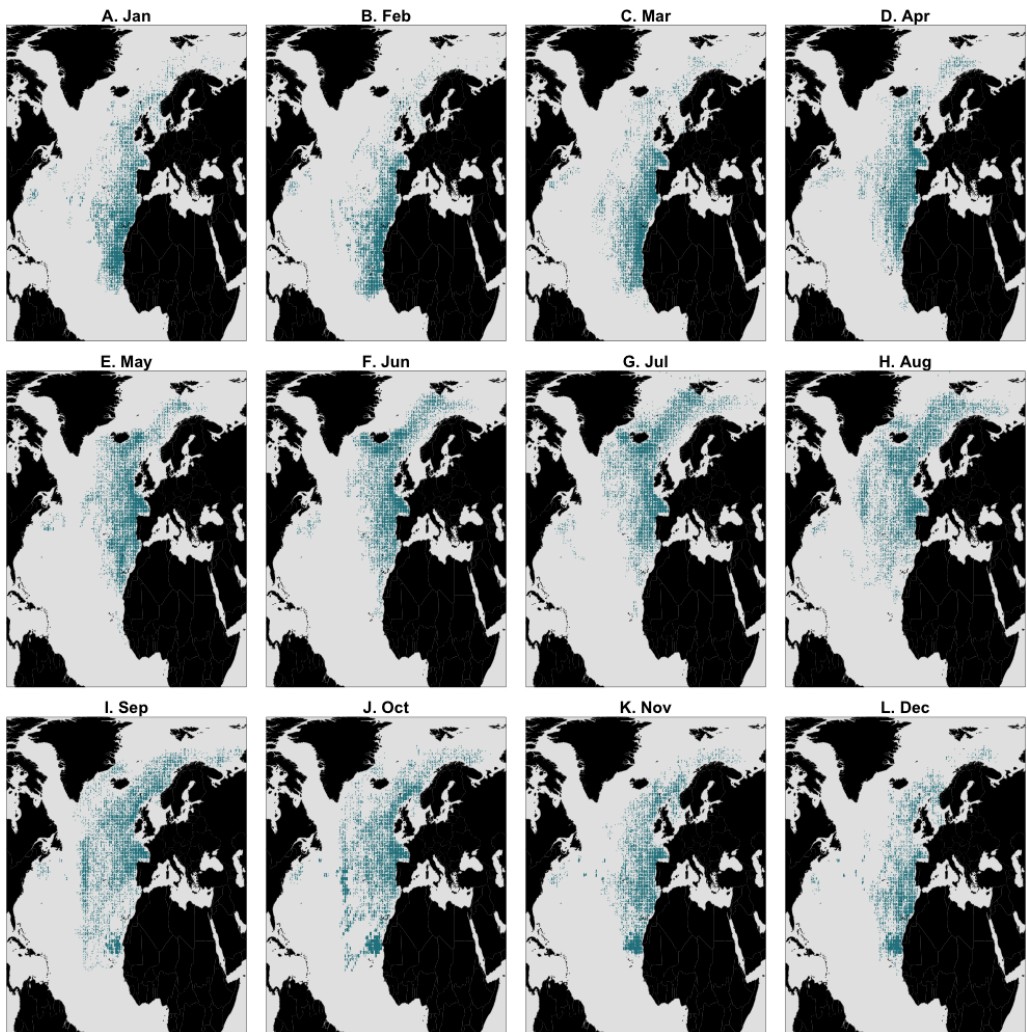

**Figure 4** Simulated locations by month taken from the top 10% best fitting migratory movement models for behavioural phase two (summer 1886–spring 1890) only.

distinct phases. The isotopic variations measured during behavioural phase one could be simulated by year-round residency in relatively warm, sub-tropical waters with limited seasonal phytoplankton blooms. The measured profile is therefore consistent with residency in warm sub-tropical waters for at least one full year.

The measured isotopic variations seen in behavioural phase two, characterized by repeated high amplitude cycles in $\delta^{13}C$ values, were best matched by movement models coded to produce annual latitudinal migrations, with best fitting models predicting greater latitudinal foraging ranges. We therefore infer that Hope conducted at least three annual latitudinal migration cycles.

Behavioral phase three, which covers the last 500 days of Hope's life, is difficult to simulate purely from movements within the known geographic range. Beginning in the winter of 1889/1890, the observed $\delta^{13}C$ values are relatively low, and remain constant for

c. 4–6 months, before increasing rapidly in the second half of 1890. Low $\delta^{13}$C values are found in northern waters, but these areas also show large temporal fluctuations in $\delta^{13}$C$_{plk}$ values (Fig. S4) unlike Hope's observed values. We suggest that the unusual $\delta^{13}$C record in the last period of life may be associated with a change in health status such as pregnancy and calf rearing. Hope was estimated to be between 10 and 15 years old when she died (based on vertebral epiphyseal fusion; RC Sabin, pers. comm., 2017), so was therefore sexually mature during this period, so this is a possibility.

Blue whales have a 10–12 month gestation period, with calving occurring in subtropical waters, and calves are weaned after 6–7 months (*Wilson & Mittermeier, 2014*). A period of fasting associated with late pregnancy and nursing could result in release of lipid carbon reserves assimilated from northern latitudes. This would be consistent with Hope's constant, low $\delta^{13}$C values. After c. 4–6 months of low $\delta^{13}$C values, Hope has a period of relatively high $\delta^{13}$C values, implying feeding in low latitude waters consistent with warmer calving grounds, shortly followed by a final northward migration before she died. We stress that bulk protein $\delta^{13}$C values alone cannot definitively identify changes in metabolic pathways associated with fasting or pregnancy.

In summary, our combined isotopic measurements and simulations allow us to propose a movement history for the last 5–6 years of Hope's life. Behavioural phase one lasting for at least a year, most likely reflects residency in sub-tropical waters, potentially in the Cape Verde, Canary current region. Subsequently, in behavioural phase two, we infer that Hope conducted three uninterrupted annual latitudinal migrations, wintering in sub-tropical waters and moving to sub-arctic waters during late spring and summer. finally, in behavioural phase three, we tentatively suggest that Hope may have given birth in the winter of 1889/1890. Following which, we infer a period of c. 4–6 months of residency in sub-tropical waters where Hope was sustained largely from stored lipid reserves. Finally we propose that Hope had a short period of feeding in sub-tropical waters in the second half of 1890 potentially during a final northward migration and eventual stranding during the return to northern feeding grounds in early 1891.

Our approach is relatively simple and we stress that comparisons between simulated and measured datasets face several limitations. Critically, the likely accuracy of any simulation is only as good as the underlying numerical isotope model(s). In our case we used an isotopic extension to the NEMO-Medusa biogeochemical model system. Our model captures broad geographic and temporal variations in carbon isotopic compositions of phytoplankton (*Magozzi et al., 2017*), but may underestimate temporal variability associated with local features such as frontal systems, eddies or coastal processes. Furthermore the model is not parameterized for specific years: rather we draw on decadal climatological average output. It is possible that spatio-temporal variations in phytoplankton compositions simulated for the start of the 21st century do not effectively capture conditions at the end of the 19th century. However, we restrict our interpretations to broad comparisons such as whether measured data are better simulated by residency or latitudinal migrations, and at such broad scales we suspect that temporal variations in the spatio-temporal distributions of $\delta^{13}$C values are a relatively minor source of variance. It would of course be possible to run

simulations using hindcast isotope models parameterized for conditions expected in the late 19th century.

A second source of uncertainty lies in the isotopic effects of trophic transfer from phytoplankton to zooplankton prey and ultimately whale baleen. These transfer effects include temporal averaging and physiological effects associated with respiration and assimilation of proteins compared to lipids. We did not include physiological models in our simulations other than adding temporal averaging to phytoplankton $\delta^{13}$C values, and a fixed correction term between phytoplankton and baleen. The amplitude of isotopic variability varies widely between simulated and measured baleen values; with higher amplitude effects in simulated phytoplankton $\delta^{13}$C values. It is possible that isotopic dampening or signal attenuation reflects partial homogenization of $\delta^{13}$C values through physiological mixing of carbon assimilated at different times (tissue turnover) within whale body fluids. The simulation framework provides a methodology to explore hypotheses related to the effects of temporal averaging and preferential nutrient acquisition. We note, however, that in many simulation model approaches, complexity comes at the cost of transparency. In this case study we chose to limit model complexity, demonstrating the extent to which measured profiles could be reproduced using relatively simple transfer functions between simulated phytoplankton $\delta^{13}$C values and whale baleen.

Whaling was an intense pressure for blue whales during the period we are analyzing (*Reilly et al., 2008*). While we are inferring spatio-temporal distributions of just one individual, our study demonstrates a method for inferring historical distributions and movement patterns for a species that was heavily depleted and is only now returning from the brink of extinction in the north Atlantic. In the case of blue whales, the primary driver influencing seasonal movements is assumed to be the seasonally variable distribution of food resources between high and low latitudes. Although ecological drivers of movement were not the focus of this study, we expect due to metabolic requirements that blue whales have a distribution that is tightly coupled to predictable spatio-temporal differences in prey abundance on ocean basin scales (*Holdo & Roach, 2013*). We suggest that the seasonal latitudinal migrations inferred during Hope's behavioural phase two are also likely to be a common movement pattern for north Atlantic blue whales, but the inference of sustained residency in sub-tropical waters might suggest plasticity in individual movement behaviours. In future, additional isotopic records from individual blue whales, observational sightings data, and multi-year satellite tracking records will allow us to test the generality of this pattern for historical and contemporary populations. Developing an understanding of the nature and plasticity of individual-level movements across populations of large whales would improve our understanding of their population structure and could assist spatially-based management decisions affecting the conservation of endangered marine species like blue whales (*Irvine et al., 2017*).

## CONCLUSION

Temporal variations in stable isotope compositions of incrementally grown tissues offer a potentially valuable record of movement in migratory organisms, but interpreting these

profiles is extremely difficult, particularly in marine environments. Here we show how a simulation framework can be used to help interpret measured data through *in silico* experimentation. By varying parameters of agent-based movement models coupled to models predicting temporal and spatial variation in plankton carbon isotope data we identify combinations of movement behaviours producing simulated baleen isotope records that are most consistent with measured data.

Our results confirm that sequential sampling of stable isotope compositions in whale baleen, combined with simulation modeling can yield plausible inferences of individual whale movements that are consistent with assumed movement behaviours, provided our various model assumptions are true. Our movement simulation modeling removes a long-standing limitation in stable isotope ecology, and can be applied to stable isotope records from any incrementally-grown tissue (e.g., baleen, pinniped vibrissae, otoliths, vertebrae, eye lenses, teeth and long bones) to estimate individual movement behaviors over multiple years. By unlocking information contained in incrementally-grown tissues we hope that a more detailed picture of individual movement behaviour in modern and historic specimens of marine species can be developed.

## ACKNOWLEDGEMENTS

We thank CJ Somes for providing $\delta^{15}N$ POM data, Bastian Hambach and Megan Spencer at the University of Southampton SEAPORT isotope laboratory for assistance with stable isotope analyses, Andrew Yool for allowing us to use and share NEMO-MEDUSA outputs, and Sam Rossman and multiple anonymous reviewers for comments on earlier versions.

### Funding
This work was funded by the British Ecological Society (grant: 5771/6815) and the London NERC DTP (training grant: NE/L002485/1) to EJC. The funders had no role in study design, data collection and analysis, decision to publish, or preparation of the manuscript.

### Grant Disclosures
The following grant information was disclosed by the authors:
British Ecological Society: 5771/6815.
NERC: NE/L002485/1.

### Competing Interests
The authors declare there are no competing interests.

### Author Contributions
- Clive N. Trueman conceived and designed the experiments, performed the experiments, analyzed the data, prepared figures and/or tables, authored or reviewed drafts of the paper, approved the final draft.

- Andrew L. Jackson and Natalie Cooper conceived and designed the experiments, analyzed the data, prepared figures and/or tables, authored or reviewed drafts of the paper, approved the final draft.
- Katharyn S. Chadwick and Laura J. Feyrer performed the experiments, authored or reviewed drafts of the paper, approved the final draft.
- Ellen J. Coombs, Sarah Magozzi and Richard C. Sabin performed the experiments, contributed reagents/materials/analysis tools, authored or reviewed drafts of the paper, approved the final draft.

## Data Availability

Data are available from the NHM Data Portal: https://doi.org/10.5519/0093278.

Code is available on GitHub and Zenodo: https://github.com/nhcooper123/blue-whale-bes, DOI: 10.5281/zenodo.2542777.

The whale specimen (accession number: NHMUK.1892.3.1.1) is held at the Natural History Museum London.

## Supplemental Information

Supplemental information for this article can be found online at http://dx.doi.org/10.7717/peerj.7912#supplemental-information.

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
