# Peer review of "Combining simulation modeling and stable isotope analyses to reconstruct the last known movements of one of Nature’s giants"

_PeerJ, doi:10.7717/peerj.7912_

## Round 0.1 · original submission · Major Revisions

After receiving two critical, and very detailed, reviews, I am suggesting a Major Revision for this manuscript before it can be considered for publication. One reviewer suggested a rejection, but provided significant and detailed information and a track-changed draft. Working through those comments, and the comments of the second reviewer, this paper can be overhauled and considered for publication. The reviewers were excellent in provide detailed feedback and specific areas to strengthen this paper. It is clear that both found value, as do I, and both found critical areas for clarification, expansion, or reworking. I strongly suggest you work through these comments. The interpretation needs work and Reviewer 1 provides several ways to address this. I look forward to your revision.

·

Basic reporting

The basic concept is great for this study - a valuable sample and dataset - and a creative combination of tools (simulating expected isotopic patterns) to then attempt and recreate movement for the animal over many years. And with some improved clarification and explanation this will be a great manuscript. Yet in it's current form, it is not yet ready for publication, as far too many assumptions are not fully supported (or clarified) and throughout the paper, there are many inconsistencies that bring up questions about the experimental design. I do not expect a significant amount of work needs to be done in altering the study nor analysis, but rather more care needs to be given to fully and clearly convey to the reader that what has been done is sound in its design.

Experimental design

As noted in the above section, further clarification on assumptions and decisions made will greatly improve this manuscript. To help, many specific suggestions and points are detailed below. But this is not an exhaustive list. Notes and suggestions below should be considered and incorporated throughout the manuscript.

The primary sections needing attention are those related to the baleen stable isotope profiles and assumptions based on these related analyses and its application to the modeled movements. The sections relating to the agent-based movement model, while more complex, are better explained, but still require some attention. As is, the methods are not sufficiently described nor justified to support the assumptions made.

Further details provided in the next section.

Validity of the findings

With regards to the isotope profiles, the following details and notes require attention:

L178 – adjust wording to state that it was assumed that growth of the baleen was constant; the 1 citation is good to have, but this is still an assumption.

L197 – before the behavioral phases are used, as done in this line for the first time, much more explanation on what these phases are, and how they were characterized and divided is needed. It is confusing to the reader to drop this note of “behavioral phase two”, without any prior mention, introduction, of discussion of this idea. This is just one of several parts where this idea is mentioned without the full clarification it deserves. This is also a good example of where the organization of the paper needs some improvement.

L198 – similarly, more clarity and description needs to be provided on how the isotopic periodicity was identified and assigned. It is mentioned here, and references Fig S1 (which also needs improvements for clarity –see below), but without full details. It is in a much later section where the R package and details are provided – all the info on periodicity assignment should be made together. Further, it is never made clear on how the isotope data was used in the Fourier Transformation – was it just the d15N data? Just the d13C data? Both together? And critically – WHY the N or C or N+C was used needs to be explained clearly to the reader.
L346 – again, clarify what data this “strong periodic repetition” was based upon. C? N? C + N? Why?

Related to this, additional detail and clarification would greatly improve Figure S1. As is, it appears that the provided inset values presumably show the periodicity for the different analyses (N full, N phase 2, C full, C phase 2) – but this brings up more questions on why the 13.5cm = 1 year was then selected as the defining periodicity for the rest of the assumptions in the manuscript.
These comments are not to question this assumption or reasoning, but rather to emphasis that further clarification should be provided such that these assumptions and justifications are fully understood by the readers.

Additional notes on Fig. S1 – can a more detailed title for the x-axis be provided? Or at least a description given relating the “Frequency” to cm in the baleen.
Also – why is Phase 1 not shown as a stand-alone, but Phase 2 is? And what about the last section of the baleen (most recent) that is described as outside of Phase 2? As is, this figure is confusing.

L358 – To improve clarity, suggest change the wording of “84 cm from the distal end” to be consistent with the labeling of the data previously described and also presented in Fig 1. It is labeled on the x-axis as “cm from youngest sample”, so references to specific positions in this timeline should follow this direction. --> “26 cm from the gingival end” or “from newest”, or similar. Just needs to be consistent.

Related to the baseline isotope comparisons, further detail should be provided to better support how temporal isotopic fluctuation was accounted for. A primary reason why it has been difficult to model oceanographic isotopic patterns over space and time is because of the highly dynamic nature of this environment and our limited ability to capture the full spectrum of samples/observations to support the scale of data necessary to make these predictions or recreate conditions. The approach here in this paper is a good one, but requires that additional clarification and justification be provided.
For instance, L233-229 needs more detailed explanation on the temporal aspect. Why was the decadal data forced? How, specifically, did the forcing change at given latitudes and why? How was seasonality incorporated? Were time lags considered between physical conditions and biological responses? Why/why not? Etc.
This part is also partially touched upon in ~L248-258; but requires additional explanation into the rational for some key assumptions and additional citations would strengthen these positions as well. i.e. – please provide a citations for the 2-4 month turnover within krill (L256); citation for keratin (L250).


Another main point that needs clarification and improvements for consistency throughout is with regards to the terms being used to categorize the different geographic regions and associated whale behavior/foraging location. At L323-330, when the 3 modeled movement behaviors are described, the later discussions and result analyses would be improved if specific terms were defined here, and used consistently throughout – including with associated figures. Further, in strategy (a) where the residency regions are described, again, key names/terms should be assigned and then used consistently throughout and in figures. For example, Figure S4 shows 5 regions, which don’t exactly match with all the terms used in L323-326; nor in L383-390 (here, each area should be specifically listed out instead of just saying “around the British Isles” or “sub-tropical regions”, etc.)
Further examples - L401-410, and throughout most of this section where the observed baleen isotopes are compared to the model.


Related to the interpretation and assumptions made – a key point that needs to be clarified is in Fig. 3, why are the d13C scales so different? It is made clear why the absolute d13C permil values are not expected to be the same, this argument was presented well; however this mismatch in scale shown in Fig. 3 is large and requires further explanation.

Related to supporting the assumptions made – suggest presenting the permil amount of amplitudes observed when discussing the modeled movement, as well as the observed isotope patterns. (i.e. L393). This will help readers better appreciate the relative fluctuation each environment/region is expected to experience.

Fig 1 – Suggest marking the “phase” boundaries on the figure. Also, it seems that the d13C and d15N values appear negatively correlated, even in Phase 1. Please explain why this was stated as not being the case.


The rest of the assessments, comparing the baleen isotope values and the modeled movement, follow well and will be better supported with the additional clarification and justification in the earlier sections. The suggested adjustments for consistency and organization outlined above, when made throughout, will greatly improve this manuscript and make it ready for publication.



Misc. suggested edits –
L289 – suggest changing the word “Directly” to “Here,” or “Meaning”, or similar.
L296-299 – this same information was just stated above. Suggest leaving it here and deleting it in the earlier occurrence.

·

Basic reporting

Manuscript is clear and concise. Introduction provides background to set research objective into perspective. The modeling component of the methods was a bit difficult to follow. There are a lot of pieces and at times it was difficult to see how they all fit together. How do the authors combine whale isotopes, agent based model, and comparison to the d13C isoscape? For me this only became clear from reviewing the provided code. I commend authors on extensive supplemental materials, easy access to data, and transparent code.

243-245: “Confident” seems like a very strong word when assuming a relative distribution for an isoscape hasn’t changed in the last century. Historical work is difficult and always makes a lot of assumptions; however,“Confident” in a scientific publication implies a degree of certainty generally imparted from statistics. I would recommend making a stronger case for this claim based on literature or clearly noting it as an assumption.

248-251: The assumption of reflecting foraging in the “weeks” prior seems like a best guess. Claim as such or provide literature to support.

253-257: The estimated completion of turn over seems again like a best guess. Please support or claim as such. “Estimate” makes the statement seem based on something concrete.

260-264: Unclear how data is being transformed with respect to trophic level between the isoscape and baleen.

265-267: I think authors go from talking about krill to killer whales? Please clarify.

454-457: Unclear clear how statements on sexual maturity relate to interpretation of data for readers not familiar with blue whale natural history

491-492: Stocks of what were so depleted? Fish? Moved outside of what area?

I recommend using either Hope or NHM blue whale as consistently as possible.

Experimental design

I will qualify that I am not well suited to judge the adequacy of the design in relation to others in the literature. However, my concerns are as follows:

1) The model appears to have been designed backwards based on the data. The data is split into three time periods based on how the authors interpret seeming changes in the isotopic profile. This adds a subjective layer to the analysis. If the goal is to describe what this one whale did then perhaps it is acceptable however, if the goal is to extrapolate to the population these determinations are problematic. It is unclear how a researcher who wanted to use a similar technique should go about assess isotopic profiles for a larger samples size.
2) More explanation is need with respect to how the growth rate is determined. It is unclear as to why d15N values cycle annually. Additionally, given d15N is variable across the North Atlantic how is this not confounded by location especially if the whale was migratory for much of her life.
3) More justification is need for why an agent based model which makes many assumption about movement is acceptable in an analysis to determine the whale’s movement. Analysis seems circular. Additionally, how sensitive is the analysis on changes to parameters in the agent-based model?

Validity of the findings

1) I greatly respect the authors’ efforts in attempt to connect isotope data to real world animal behavior such as movement patterns. I think too often practitioners of stable isotopes cannot adequately describe what variation along their isotope axes mean ecologically. Trueman et al. provide potentially strong methods as to how isotope data may be put into a modeling framework and may be useful for specific hypothesis testing scenarios. However, I did not review this manuscript as a methods paper and would not be strongly qualified to do so. Rather I reviewed the paper based on the ability to achieve the stated objective in lines 83-87: “Here we illustrate the potential of this conceptual approach by assessing the relative likelihood of differing, plausible hypotheses of movement behavior for a single individual blue whale based on carbon isotope compositions sampled across a baleen plate” as well as the title “Reconstructing the last known movements of one of Nature's giants”. Based on this, it is I find that the authors did not achieve the stated objective nor were the movements of the blue whale in question reconstructed. In science hypotheses must be falsifiable. For the last period of the whale’s life the observed pattern of isotope values could not be simulated. It did not match the pattern that would be expected for low latitude residency, yet the authors interpret the pattern as low latitude residency and explain the anomaly as a possible pregnancy. While based on my experience in stable isotope ecology, I think this sounds plausible, the fact remains that when data was encounter that did not support the hypothesis that the whale had an extended low latitude residency, the hypothesis was not rejected. Thus, it fails to be falsifiable. Additionally, it leaves me asking: under what scenario would the authors have deemed this conceptual approach incapable of assessing the relative likelihood of differing, plausible hypotheses of movement behavior?
2) Trueman et al. is attempting something extremely difficult as it relies on biological processes which span all the way from the incorporation of carbon into the biomass of primary production at the base of the food web to the synthesis of keratin by literally the largest known animal to have ever existed. These are complex processes and variation is inherent. For this reason, the model must stack uncertainty upon uncertainty: how accurate is the isoscape? how accurate is the agent-based model? How accurate is the estimated rate of keratin growth? And finally, how sensitive are results to variation in all of these underlying estimates? As none of these are discussed it is really difficult to interpret how potentially useful this work is.

Along these lines:
Authors claim the Suess effect and trophic fractionation do not impact results as they only use the change in d13C not the absolute value. However, this is only true if we assume the Suess effect and trophic fraction are uniformly impacting the entire study region. The Suess effect impacts low latitudes far more than high latitudes given the much larger pool of dissolved CO2 in cold water. Additionally, trophic enrichment may be variable based on the growth rate of the organism (among other things) which may be related to productivity. I agree that these are unlikely to impact results however I think assumptions (especially for modeling heavy papers) should always be transparent.

The modeled and observed data for Magozzi model are most desperate off the west coast of northern Africa and in polar regions (but in opposite directions) given that these are both important locations for blue whale how might that impact the authors’ ability to model movement?

Lines 505-509: How can you make inferences about what is common in modern blue whales from a single historical sample? This idea could be flush out a bit more and should be noted as a speculation or prediction.

Lastly, in lines 528 to 532 when authors claim: “Our results confirm that sequential sampling of stable isotope compositions in whale baleen, combined with simulation modeling can yield plausible inferences of individual whale movements that are consistent with assumed movement behaviours.” This is only true if the large number of assumptions inherent in their analysis are also true. Authors need to be more explicit about assumptions and potential violations.

Additional comments

I commend the authors for this ambitious analysis. I appreciate the amount of time and effort that was clearly placed into this complex analysis. I struggled as to how to interpret the paper, as either methods paper or scholarly research. It didn’t seem like a methods paper as the focus was clearly the blue whale and there wasn’t any sort of model validation. If the authors deem the methods in the paper sufficiently novel, I would encourage them to focus more on the methods and use the whale as a proof of concept. As it stands, the samples size of one makes this a case study not a research article. Given the detailed code the authors make available I would encourage them to really think about the circumstances in which a potential researcher should or should not use these methods. As I am sure authors are aware new quantitative techniques are emerging using stable isotopes that have awesome potential to answer novel questions and provide estimates of things like diet that were previously only obtainable through substantial effort. However, given the nuance of isotope analysis and propensity for researchers to preform analysis they don’t completely understand I would encourage authors to provide some guidance as to the circumstances in which this sort of analysis would or would not be appropriate.

A few additional comments:

Please provide any additional details as to the circumstances of the stranding, could the whale have been sick/injured etc? The details of the stranding could inform interpretation of last behavioral phase.

187: Remove per mil from C:N ratios

Was the number of simulations limited due to computing constraints? Perhaps justify the number of simulations. 1059 doesn’t seem like a large number given the potential number location histories.

See other line item corrections in attached PDF

Sincerely,
Sam Rossman

---

## Round 0.2 · Minor Revisions

Thank you for a wonderful effort to meet the comments from the reviewers directly. Both reviewers that re-reviewed the article feel the manuscript is publishable. I have left this in Minor Revision to give you an opportunity to review the minor comments from one of the reviewers. Once those are made, I will happily accept the manuscript for publication. Thank you for your efforts and attention to the comments. Great work.

·

Basic reporting

The re-framing of this MS made a significant improvement. It is now not only a more accurately focused paper, but it is now an extremely useful guide for future studies that hope to utilize SI data and model simulations to infer movement and behavior of migratory marine animals.

Other incrementally grown tissues that are also comparable to this type of study include teeth and bones (i.e. long bones such as humeri); if not directly discussed, both "teeth and bones" should be added to Line 615 at the end.

Experimental design

Good adjustments and added clarifications. The new set up which presents a binary test of expected behavior and associated SI pattern (residency vs seasonal migration) makes this a much more focused study with a clearly testable and realistic hypothesis. The more simplistic approach taken by limiting model complexity is a great adjustment.

Validity of the findings

Much improved after more clearly defining and supporting the assumptions made and process applied.

Additional comments

Overall - the adjustments made greatly improved this MS. Specific line edits/comments are listed below - and the only main overarching comment is to given attention in the Results and Discussion section to clarify, at a final level, the 3 behavioral phases that are identified in the baleen sample and then compared to the simulated results. There is a bit of jumping around during the Results and Discussion section that makes it a bit disjointed, but with a bit of additional clarifying wording, this will be easily resolved. You'll see this theme in many of the comments below.
Again - great refocusing, and this will be a great addition to the spatial / behaviourial stable isotope literature and field.


Misc. suggested edits –
L 25 – should be “relative” not “relatively”
L 446 - Suggest adding one line at the start of this paragraph to describe what is being done here. i.e. “For each of the three behavioral phases identified within the baleen plate, the modeled simulation results were compared.” (or similar)
L 457 – add “likely”; “…the observed low δ^13C values likely imply foraging in colder, more northerly latitudes…”
L 460 – to further clarify the connection between these two paragraphs, suggest adding this phasing to the start of L460, “To further examine this, we modeled…
L 461 – deleted the repeated phrase “compared the”
L 480 – this section seems to move through the initial assessment of phase 1 and 2 – but it seems odd that the Discussion section begins without mention of phase 3. Suggest adding a similar descriptions/assessment of phase 3 here, before moving on to the Discussion section.
Adjust the Discussion, then, if/as necessary to eliminate any repetitiveness.
L 494 – says “two distinct phases” – but wasn’t this adjusted to 3 phases? Clarify.
L 507 – this is now describing “phase 3” right? If so, state this here to clarify.
L 528- Here, and overall, there needs to be adjustments made to make the comparisons and discussion of the 3 behavior phrases consistent throughout. Suggest making it explicitly clear to the reader which phase is being discussed/mentioned each time.
i.e. “In summary, our combined isotopic measurements and simulations allow us to propose a movement history for the last 5-6 years of Hope’s life. Behavioural phase one lasting for at least a year most likely reflects residency in sub-tropical waters, potentially in the Cape Verde, Canary current region. Subsequently, behavioural phase two, we infer that Hope conducted three uninterrupted annual latitudinal migrations, wintering in sub-tropical waters and moving to sub-arctic waters during late spring and summer. Finally, in behavioural phase three, we tentatively suggest that Hope may have given birth in the winter of 1889/1890….” Or similar.
L 552 – add a comma after “However”
L 615 – should add teeth and bones (or long bones) to this list

·

Basic reporting

Very professional and concise. Flow of ideas is easy to follow and logical. I think the headings provide a useful guide for readers and aid in the clarity of the manuscript.

Experimental design

The details relating to the exp. design, assumptions and limitations of the work are now well described. It's not that I thought the assumptions in the previous version were unreasonable, I just felt that the needed to be more explicitly stated with the possible implications. Authors do a good job of this in the current version.

In my previous review my recommendation of reject was not really based on the merit, usefulness or quality of the research but rather that I felt it was mostly a case study which is not within the scope of the journal. I think author's have sufficiently reframed the paper with a more methodological focus.

Validity of the findings

The authors do a much better job in this version of separating the which parts of the results are well supported by their analysis and which are more speculative. While the results of this case study are not, by themselves, a huge contribution to the literature given the single individual it is a valuable approach which if adopted by other marine mammologists has to potentially to shed light on threatened and hard-to-study populations.

Additional comments

I commend the authors for a dramatically improved manuscript. I felt the response to previous comments were thoughtful and well articulated. I feel the manuscript is now very clear, and much easier to follow. In it's current form I believe the manuscript will be a far more useful contribution to the literature as the accessibility of the research has been dramatically improved for researchers less familiar with these sorts of models. Additionally, framing the paper around evaluating support for specific hypothesis turns the focus of the work from something I think is mostly impossible (constructing fairly specific movement patterns) to something useful and very applicable to other organisms.

---

## Round 0.3 · accepted · Accept

Congratulations! Your manuscript is acceptable. Thank you for the full faith effort to work through the revisions on this latest draft. The paper is ready for publication in PeerJ. Well done.